# Psychiatric Partial Hospitalization Programs: Following World Health Organization Guidelines with a Special Focus on Women with Delusional Disorder



**Alexandre González-Rodríguez** [1,*], **Aida Alvarez** [2], **Armand Guàrdia** [3], **Rafael Penadés** [4], **José Antonio Monreal** [5], **Diego J. Palao** [1], **Javier Labad** [6] and **Mary V. Seeman** [7]

1   Department of Mental Health, Institut d'Investigació i Innovació Parc Taulí (I3PT), Parc Tauli University Hospital, Autonomous University of Barcelona (UAB), CIBERSAM, Sabadell, 08280 Barcelona, Spain; dpalao@tauli.cat
2   Department of Mental Health, Institut d'Investigació i Innovació Parc Taulí (I3PT), Parc Tauli University Hospital, Sabadell, 08280 Barcelona, Spain; aalvarezp@tauli.cat
3   Department of Mental Health, Parc Tauli University Hospital, Sabadell, 08280 Barcelona, Spain; aguardia@tauli.cat
4   Barcelona Clinic Schizophrenia Unit (BCSU), Neurosciences Institute, Hospital Clinic of Barcelona, University of Barcelona (UB), IDIBAPS, CIBERSAM, 08029 Barcelona, Spain; RPENADES@clinic.cat
5   Department of Mental Health, Hospital Universitari Mútua de Terrassa, Autonomous University of Barcelona (UAB), CIBERSAM, 08221 Terrassa, Spain; jamonreal@mutuaterrassa.cat
6   Department of Mental Health, Consorci Sanitari del Maresme, Fundació Parc Taulí, CIBERSAM, 08304 Mataró, Spain; jlabad@csdm.cat
7   Department of Psychiatry, University of Toronto, Toronto, ON M5P 3L6, Canada; mary.seeman@utoronto.ca
*   Correspondence: agonzalezro@tauli.cat; Tel.: +34-937-458-377

**Abstract:** The World Health Organization (WHO) developed a 7-year Mental Health Action Plan in 2013, which recommends integration of health and social care services into community-based settings, implementation of strategies for health promotion and prevention of illness, and support of research. In this review, we highlight partial hospitalization programs (PHPs) for delusional disorder (DD), with a special focus on the health and psychosocial needs of women. We suggest that PHPs are, in many ways, ideal settings for carrying out WHO recommendations. PHPs are multidisciplinary and consequently are able to provide a wide range of flexible program offerings. Programming in PHPs is able to address, with proven efficacy, individual needs, such as those presented by women at the various stages of their reproductive life. PHPs are a community bridge between hospital and outpatient services and can quickly adapt to specific needs as affected by gender, but also by age and cultural origins. They are ideal settings for professional training and for conducting clinical research. PHPs operate on the principle of shared decision making, and thus more readily than many other treatment sites, engaging difficult-to-treat patients, such as those with DD, by successfully establishing long-term relationships of trust.

**Keywords:** psychosis; delusional disorder; partial hospitalization; community settings; gender-specific treatment

## 1. Introduction: Delusional Disorder

For many decades, delusional disorder (DD) has been perceived by health professionals as a treatment-resistant disorder [1]. DD is also reported as being problematic for family members, partly because DD patients adhere poorly to prescribed medications, do not willingly attend treatment appointments, and seem often to be unaware that they are ill [2]. It is, however, possible to help families develop home-based therapeutic skills while patients receive specialized treatment in the community [3]. The current *Diagnostic and Statistical Manual of Mental Disorders*, Fifth Edition (DSM-5), defines DD as a disorder characterized by the presence of one or more delusions lasting for at least one month, without

prominent hallucinations, affective symptoms or other psychotic symptoms such as those seen in schizophrenia [4]. Despite this definition, research into the disorder points to a significant association with affective morbidity [5], seen more frequently in women than in men. This is one reason why gender-specific interventions are important in psychiatric services for DD.

Epidemiological data regarding DD are not without controversy. DSM-5 reports a lifetime prevalence of approximately 0.2%, whereas some publications have estimated the prevalence of DD to lie between 24 and 30 per 100,000 [6]. Psychosocial risk factors and research settings influence the prevalence and incidence data. As an example, studies of prison inmates show an estimated prevalence of nearly 0.24%, substantially higher than that found among community recruits [7].

Multidetermined gender differences in psychopathological and clinical features of schizophrenia, a disorder related to DD, have been widely investigated, but are underresearched in DD [1]. DSM-5 mentions no gender differences in the content of delusions or in the prevalence of DD, whereas some research has reported significant gender differences in both affective and substance use comorbidity [8]. Discrepancies among studies are probably due to the major influences of age, reproductive status, and cultural influences on this disorder. Importantly, DD is a psychotic disorder that typically starts in middle age, which, in women, coincides with menopause, the end of the reproductive life cycle [1]. This is a time of increased vulnerability in women, accompanied by specific medical and psychosocial needs [9]. Characteristically, age at onset of DD is approximately 45 years, which means a physiological decline in estrogen levels, an increase in medical comorbidities, and psychological losses for women—loss of fertility, adolescent children leaving home, and aging parents dying [1]. Such factors, as well as cultural traditions and the impact of socio-economic circumstances in women, influence the presentation of DD and determine best practice.

Our research group conducted a retrospective longitudinal study with a one-year follow-up, which included 78 patients with DD consecutively admitted to an inpatient unit [10]. When compared to men, women showed a delayed age at onset and required a longer duration of hospitalization. However, other investigators studying outpatient populations have found no gender differences in the course of illness or resource need [11]. Partial hospitalization patients have not been studied.

We undertook the present review in the context of the 2013–2020 World Health Organization (WHO) goals for mental health, which WHO characterized as being fundamental to global health [12]. WHO stated that persons with mental illness currently suffer service discrimination and neglect. The organization elaborated a mental health action plan that emphasizes the integration of health and social care services into community-based settings. The plan includes the implementation of strategies for health promotion and prevention of illness, as well as advocating for an increase in the research evidence base [12]. WHO also addresses women's human rights and recommends gender-responsive approaches that recognize the need to protect women in situations of psychosocial risk, such as barriers to education, migration stress, and domestic abuse.

Thus far, very few studies have focused on women's needs in the specific context of DD [13]. Since all women with psychosis had been reported to engage poorly with physical health services, our group recently investigated the rates of gynecological service use in outpatient women with DD [14]. We found that 48% of the sample had not received gynecological attention over the past 2–3 years. Because this population showed poor adherence to gynecological screening and low attendance at gynecological appointments, we recommended a potential solution—the reinforcement of the nurse–patient relationship. This relationship is considered to be crucial for patient engagement, attendance at appointments, adherence to prescription drug regimens, and compliance with routine health screening [15].

The main goal of this paper is to review how the 2013–2020 WHO recommendations apply to partial hospitalization programs for DD, with a special focus on women patients. Specific aims of this review are: (1) to describe a community treatment approach for patients

with DD that is in line with WHO recommendations and (2) to describe how the WHO recommendations for the treatment of severe mental illness in women apply to women with DD. This includes prevention of medical and psychiatric comorbidities and the promotion of anti-stigma campaigns. Finally, we propose a model of partial hospitalization (mainly day hospital) that is compatible with WHO goals and easily adaptable to the specific health and social needs of women.

## 2. Methods

This is a narrative non-systematic review based on searches in the PubMed database for English, German, French and Spanish language papers published from inception until March 2021 and referencing community-based mental health treatment models (including partial hospitalization programs (PHP) for patients with psychosis (which includes DD)). Google Scholar was also searched to find other relevant papers in the field, not available on the PubMed database. Reference lists of retrieved papers were also searched.

The review addresses WHO's Mental Health Action Plan 2013–2020 and discusses its applicability to women with DD attending a PHP. In the conclusion, we propose targets and actions that need to be developed in PHP programs and that can add to the evidence base of interventions that benefit women with DD.

We started with searching the following key terms: ("partial hospitalization" OR "partial hospitalization programs" OR "day hospital" OR "day hospital care" OR "community mental health") AND ("delusional disorder" OR psychosis). We then searched WHO's Mental Health Action Plan [12] and, relying on the recommendations found in that document, we proposed objectives and functions of PHP as sites of treatment for women with DD. This section provides the results and conclusions that can be drawn from the available literature.

The screening and selection processes were undertaken by AGR and AA. Several hundred titles and abstracts were scanned and selected if they were judged relevant to the treatment of DD in community-based mental health services or partial hospitalization programs. Figure 1 shows the procedure we undertook for the selection of papers. Ultimately, a total of 95 papers were selected (Figure 1).

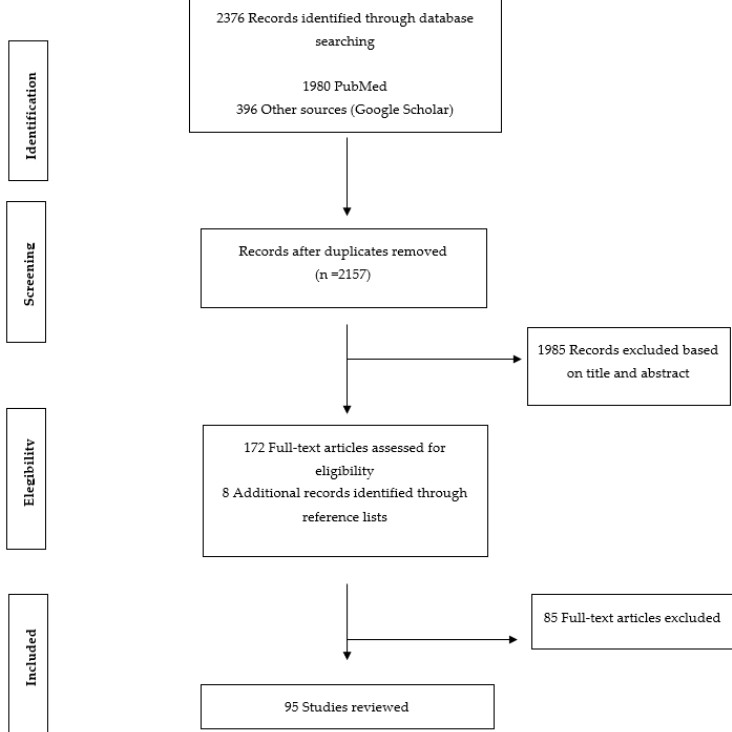

**Figure 1.** Flowchart for inclusion of studies.

### 3. Community-Based Mental Health Models in the Treatment of Delusional Disorder

The evidence available to date suggests that a mental health model centered on in-hospital provision of care is insufficient for community needs [16]. A combination of hospital and community care is required for best practice [17]. We define community mental health care as founded on the principles and practices necessary to promote the mental health of a population belonging to a specific geographic area. Its objectives are to:

(a)   address the needs of a specific population, with accessibility viewed as a crucial element;
(b)   harness the goals and strengths of persons with mental illness through rehabilitation, leading to recovery;
(c)   promote a network of adequate support, services, and resources via multidisciplinary teamwork;
(d)   emphasize services that are evidence-based and recovery-oriented [18].

Community mental health care is organized around two fundamental pillars: (a) mental health in primary care and (b) community mental health centers (CMHC). Providing mental health services in primary health care involves assessing, diagnosing, and treating people with mental disorders, putting in place strategies to prevent mental disorders, and ensuring that appropriately trained primary health care workers are able to provide the necessary psychosocial and behavioral therapeutics. Required tasks include sensitive interviewing, counseling, and making use of interpersonal skills in order to improve overall health outcomes [19], as well as facilitating, in complex cases, access to specialized community health centers (CMHC).

The objective of CMHCs is to ensure that people with serious mental health problems are treated within their community, reducing the need for hospital admission as much as possible and favoring integration into society. These centers are staffed by multidisciplinary teams that include psychiatrists, nurses, social workers, psychologists, and occupational therapists working as an integrated team. The functions of CMHCs are summarized in Table 1.

**Table 1.** Functions of community mental health centers [20].

| Functions of the CMHC |
| :---: |
| Providing biological and psychosocial interventions for individuals with severe mental health problems |
| Cooperating closely with primary health care and hospital units |
| Providing psychoeducation and focused support to families |
| Coordinating with other institutions and community service organizations |
| Sustaining social functioning and increasing the time individuals spend in the community |
| Contributing to the reduction of mental illness stigma |

One form of community mental health outreach is Assertive Community Treatment (ACT). ACT is a mental health team treatment modality shown to be effective in adult patients (including men and women with DD) with severe mental disorders (SMIs). ACT provides treatment in patients' homes and encourages peer counseling [21].

#### 3.1. The Role of Community Treatment in Delusional Disorder

Delusional disorder (DD) is less prevalent than other serious mental disorders, i.e., 0.02% compared to 1% for schizophrenia and bipolar disorder [22]. Patients with DD are also better able to function in society than other patients with a serious form of chronic psychosis, which often means that they do not seek treatment unless coerced by family or by a judge's order. In most cases, DD patients do not ask for help for delusional thinking (which they do not recognize as delusional) but rather for the anxiety, insomnia, or medical complications that the delusions provoke [23]. For this reason, community identification and appropriate referral become all the more important.

### 3.1.1. Primary Care in Delusional Disorder

The importance of early recognition of psychotic disorders has been underscored by studies suggesting a possible association between duration of untreated psychosis and eventual outcome [24,25]. Due to the relative rarity of DD, general practitioners are not experienced in recognizing these disorders, but the development of screening tools will help. Verdoux et al. [26] conducted a study to assess the prevalence of delusional ideation in primary care patients and the feasibility of screening. Eighty percent of the patients completed the screen willingly, while the 20% of those who refused were subsequently diagnosed with delusional symptoms. There are few data on the precise prevalence of delusional disorders among primary care patients, but Verhaak [27] reported a 2–4% prevalence among patients whose presenting complaint was social or psychological. Given these findings, facilitating the detection of DD in primary care patients appears feasible and of benefit to society.

### 3.1.2. Community Mental Health Centers and DD

Community interventions in DD and other psychotic disorders have been shown to be as, if not more, effective as the older models of institutionalization. Among the main benefits is a significant reduction in relapses and hospital readmissions, pointing to improved adherence to follow-up care and pharmacological treatment. User satisfaction and global functioning are also improved. Among the usual services offered are [28]:

(1)  Rapid identification of emergencies and crises, thus avoiding decompensation and hospital admission.
(2)  Availability of psychological therapy, such as cognitive–behavioral, systemic family, group, and occupational therapy, all of which improve insight and prevent relapse, readmission, and secondary symptoms such as anxiety.
(3)  Ready access and support to family members.
(4)  Rehabilitation and supportive work opportunities.

In conclusion, evidence-based interventions in the community setting offer comprehensive treatment for DD and related psychotic disorders, favoring autonomy and helping to achieve occupational and relational goals, thus improving quality of life.

### 3.2. The Concept of Empowerment in DD

Springing from developments in community care and psychosocial rehabilitation, the concept of "empowerment" currently reflects a new approach to mental illness [29–31]. In contrast to traditional medical models, more decision-making power is now placed in the hands of patients who decide on their own goals and are supported in practicing self-management techniques that not only reduce symptoms but also allow achievement of personal life aims [32].

The WHO European Regional Office [33] defines empowerment as "a multidimensional process through which individuals and groups acquire better knowledge and control over their lives. As a consequence, they can change their political and social environment to improve their health-related life circumstances." The term refers to a rebalancing of power among the main stakeholders in the care process: professionals, patients, and family members.

Considering the fact that patients with DD are, on the whole, well-functioning, the concept of "empowerment" is very pertinent to their treatment. There is growing evidence that this principle is not only popular and equitable but also more effective than a strictly medical model of treatment. The empowerment model of recovery improves care, increases treatment adherence, and helps to prevent relapse [34]. Recommendations on ways to support empowerment are as follows:

(1)  Maintain an attitude of respect and concern for the health and dignity of people at all times [35,36].

(2) Promote patients' personal self-help and support processes through their social networks [36].

(3) Support the development of practical problem-solving and decision-making with respect to treatment and other aspects of life [36].

(4) Establish a non-paternalistic relationship framework that favors consensus [37].

(5) Offer sufficient information to enable patients to make knowledgeable decisions [36].

These relatively new concepts in psychiatry correspond to the objectives and strategies of rehabilitation. In addition, they reflect the wishes of the majority of the population and are in concordance with the WHO action plan.

With regard to the relationship between community health models and partial hospitalization programs, Evan and collaborators have described the conversion of partial hospitalization to recovery-oriented programming in the context of health system reform [38]. The Evans et al. paper explains that PHPs were initially oriented to medication management, group treatment and activity groups. Patients initially referred to PHPs stayed for prolonged periods; the vast majority were heavily medicated and needed intensive rehabilitation. However, once they were based in the community, PHPs became sites of recovery [38]. Other medical specialties have applied a similar model of partial hospitalization by including family-based integrated care among the treatments provided [39,40]. Thus far, discussions on partial hospitalization in mental health have remained focused on the biomedical and psychosocial paradigms of recovery [41] without specific attention paid to empowerment concepts as they apply to subgroups of patients such as women.

## 4. Core Strategies of the Mental Health Action Plans Designed for Patients with Delusional Disorder

In the following section, we select the core strategies for mental well-being found in the WHO Mental Health Action Plan (2013–2020) that are applicable to patients with DD [12]. These include promotion of mental health and well-being and prevention of mental illness as well as the adoption of a well-integrated social and medical model of care (Table 2).

**Table 2.** Strategic WHO objectives and proposed actions as they apply to DD.

| Theme 1 |
|---|
| Promotion of mental health |
| Prevention of mental health |
| **Theme 2** |
| Improvement of quality, equity, and continuity of care |
| **Theme 3** |
| Integration of health and social models of care |
| **Theme 4** |
| Appropriate training for health workers |

### 4.1. Theme 1: Promotion of Mental Health and Prevention in Delusional Disorder

WHO is convinced that providing care only after symptoms emerge is insufficient. WHO supports early prevention [12], which includes improvement in the social and economic circumstances of the citizenry and attention to their physical health needs, education, employment, and adequacy of income while ensuring destigmatization and protecting against adverse effects of treatment [12]. It is well known that mental disorders are a major cause of unemployment and poverty [42]. Workplace interventions and reintegration is therefore critical, especially in patients with DD who are, for the most part, cognitively intact and able to work but whose employment may suffer because of interpersonal difficulties resulting from paranoid thinking [43].

WHO is also concerned with the prevention of suicide [12]. Its initial target was to reduce suicide rates across the world by 10% by 2020. Despite some studies (whose samples include DD patients) reporting a 15% lifetime attempted suicide rate in this condition, definitive data on completed suicide in DD are lacking [44]. The advent of COVID-19 in 2020 unfortunately increased suicide rates across all disorders [45,46], so that the WHO goal could not be reached. Suicide prevention continues, therefore, to be a critical goal for the future. The reduction of substance abuse, another goal stymied by COVID-19 [47], also needs to be postponed. In DD, substance use disorders are less frequent than in other psychotic disorders but, when present, are less frequent in women than in men (though women exhibit negative effects more rapidly than men) [8].

### 4.1.1. Objective 1: Promotion of Mental Health and Prevention of Mental Disorders

Education, psychosocial support, and employment are known to be protective against DD [48]. Pillmann and collaborators hypothesized that reactive DD (cases where there is an obvious precipitating factor) show a better prognosis than cases without triggers [43]. Nineteen patients with DD who experienced a stressful life event prior to the onset of the illness were compared with 24 patients who had not. Patients with reactive DD tended to be younger, but no clinical course differences were found. Despite the negative results, employment interventions have face validity and have been recommended [33,48].

### 4.1.2. Objective 2: Prevention of Suicidality and Substance Use

The frequency of suicidal behavior in DD has been estimated at 8–21% [49]; several sociodemographic and clinical features have been associated with increased suicide risk [45]. Severe stress is a frequent precursor [50], perhaps especially for patients with somatic delusions who often suffer high rates of comorbid depression. The early identification of comorbid depressive symptoms in DD is reportedly an effective target in the prevention of suicide. Women, as is well known, show more frequent depression than men [8]. The WHO guidelines endorse depression treatment as critically important in the reduction of suicide [50].

Substance abuse has long been associated with an increased risk of suicide [51]. The path from substance abuse to suicide may be mediated through depression.

### 4.1.3. Objective 3: Anti-Stigma and Discrimination Actions

Anti-stigma and anti-discrimination actions occupy a prominent place in the comprehensive action plan of WHO 2013-2020 [12]. Promotion of mental health includes abolishing stigmatization and discrimination. WHO recommends strategies such as building trust and improving working conditions aimed at combatting stigmatization, discrimination, and other human rights violations against persons with mental disorders [52]. Reducing stigma and addressing public, institutionalized, and internalized stigma will, it is hoped, overcome major barriers to seeking help in men and women with DD [53], especially perhaps in vulnerable populations, such as seniors, new immigrants, pregnant women, seniors, and ethnic minorities.

### 4.2. Theme 2: Improvement of Quality, Equity, and Continuity of Care

Resource planning is crucial to service improvement. WHO recommends that mental and physical care needs be integrated [12]. In the context of DD, it has been shown that women do not follow through on their medical and surgical referrals [15]. Closer liaison between psychiatry and other medical specialties will help to overcome current barriers.

### Objective 4: Improving Quality, Equity, and Continuity of Care (Including Physical Health) in DD

One of the global objectives of WHO is to provide comprehensive, integrated and responsive mental health and social care services in community-based settings. The WHO document sets a target of 20% increase in service provision [12]. It proposes stepped care

principles—primary care, community-mental health services, short-stay inpatient care, and outpatient care in general hospitals. The role of day care centers for persons with mental disorders is also discussed, but in DD, there has been, thus far, little solid evidence for the effectiveness of partial hospitalization programs.

*4.3. Theme 3: Coordination and Cooperation across Different Levels of Care (Measuring Health and Social Outcomes)*

WHO's goal is "to provide comprehensive, integrated and responsive mental health and social care services in community-based settings." This objective means increased coordination and cooperation across different levels of care [12].

With respect to DD, the mission is difficult because individuals suffering from DD tend to avoid mental health services, not considering themselves psychiatrically ill. Basic trust needs to be established by doing more listening than talking when assessing patients, and treating patents, no matter how severely delusional, with respect [3]. A focus on the needs of the different stages of the life course is helpful, and especially so in women because their needs differ across reproductive stages. The onset of DD in women with DD coincides with menopause when many medical and psychosocial problems also emerge and demand attention [54].

*4.4. Theme 4: Specific Training for Mental Health Professionals*

The identification of gaps, specific needs and training requirements for health workers is strongly recommended by WHO. The document stresses the need for training that is culturally appropriate to patients [12]. Immigrants to a new country show a relatively high rate of delusional disorders, which may take the form of culture-specific syndromes, some of which are sex-specific [55]. Emigration stress, as well as the process of migration itself, plus assimilation stress after immigration, all contribute. In order to appreciate the many problems associated with immigration, therapists require special training.

## 5. Partial Hospitalization Programs for Women with Delusional Disorder: Targets and Actions

A PHP is a service program that bridges the gap between hospital and community mental health settings [56]. Patients live at home and attend the PHP up to seven days a week to receive family and group therapy, psycho-educational programming, individual therapies and psychopharmacological and cognitive assessment and monitoring [57]. Staff is multidisciplinary and can offer patients additional programs such as exercise programs, budgeting help, school and vocational help, art and music therapies, and other creative outlets.

As per the recommendations of WHO, health promotion, disease prevention, and attention to physical health can all be delivered in the PHP setting.

*5.1. Health Promotion in Women with DD*

With respect to women with DD, PHPs provide specific preventive measures and therapeutic interventions for physical as well as psychosocial health. Psychologists and psychiatric nurses play an important role in these tasks. Table 3 summarizes some of the potential pro-women actions that can be taken by a multidisciplinary staff in the context of PHPs.

Medical comorbidities, often sexually dimorphic, are frequent in patients with psychosis (including DD) [58]. As an example, our group has recently reviewed the importance of cancer prevention and care in men and women with psychosis [59].

Rates of smoking have declined in the general population but not among people with psychosis [60]. Smoking cessation therapy and the prevention and treatment of unhealthy lifestyles are usually driven by psychologists and trained nurses in PHPs [61,62]. Many different therapeutic modalities are used in smoking cessation programs, such as motivational interviewing, cognitive–behavioral therapy for smoking as well as the use of specialized mobile apps. These are effective once trust is established and psychoeducation

and support are provided. Pharmacological strategies that help in smoking cessation can be used, but cautiously, because they may exacerbate psychotic symptoms [61].

**Table 3.** Role of psychiatric nurses and psychologists in PHP for health promotion and prevention of mental and physical ill-health in women with DD.

| **Health Promotion and Prevention of Mental and Physical Ill-Health** | |
| --- | --- |
| **Nurses** | |
| Individual assessment and therapy | Medication management<br>Psychopathological monitoring<br>Health promotion (cancer screening and adherence to psychiatric, medical, and gynecological appointments)<br>Patient and family support |
| Group therapies | Lifestyle intervention: motivation, learning, empowerment strategies, and behavioral skills<br>Psychoeducation<br>Sleep hygiene |
| **Psychologists** | |
| Individual assessment and therapy | Psychopathological assessment and behavioral monitoring<br>Identification of psychotic exacerbation and emotional responses to delusional ideas<br>Encouraging adherence<br>Cognitive–behavioral therapy for insomnia, smoking cessation, and psychotic symptoms |
| Group therapies | CBT for psychopathological symptoms<br>CBT for insomnia and smoking cessation<br>Psychoeducation<br>Family therapy |

Psychiatric nurses play a pivotal role in all mental health care settings and this is true for PHPs [62]. The assessment and maintenance of patients' safety is carried out by psychiatric nurses. Individual therapy by nurses is focused on medication management, assessment of psychopathological symptoms and behavioral disturbances, health promotion (mental and physical health), and patient and family support [63]. The latter is crucial to help family members interact therapeutically with patients, ensure adherence to treatment, and develop home strategies to reduce tension and prevent behavioral disturbances. Nurses, who are most often female, are important in the care of women with DD, especially with respect to counseling around interpersonal problems, parenting issues, and child alienation, a frequent occurrence in DD. Nurses are knowledgeable about contraceptive methods and menopausal complaints and skilled at ensuring regular cancer screening for their patients as well as regular attendance at medical, dental, and gynecological office visits.

Delusional parasitosis, one form of DD, is frequently observed in patients with physical disorders. Ozten and colleagues [64] reported the case of a 70-year-old woman with delusional parasitosis and comorbid hyperthyroidism. They recommend thyroid function tests for all patients when they receive a first diagnosis of psychosis—this is especially important in women in whom thyroid disease is much more prevalent than it is in men. Medical tests and medical referrals are easily done in PHPs. Lifestyle disorders are important targets for treatment by PHP psychologists. A recent review described advances in psychological interventions for type 2 diabetes and cardiovascular disorders [65]. Self-management techniques, sleep hygiene [66], and educational interventions based on learning and motivation are behavioral skills that can be taught to both men and women in PHPs.

Khazaie and collaborators [67] reported the case of a 60-year-old woman admitted to an inpatient service with complaints of insomnia, agitation, and suicidal ideation. She was diagnosed with DD manifesting as sleep state misperception that responded to an antipsychotic, olanzapine. This particular case, because it centered on sleep, would have been difficult to diagnose and effectively treat in a day hospital, but partial hospitalization at night, though rarely offered, is also a possibility, especially valuable for patients who work full time.

While patients with DD are often difficult to treat [3], the PHP setting has advantages in that patients continue to be in daily contact with family members and neighbors so that common triggers of acute episodes (arguments, obligations, and expectations experienced as excessive, unexplained noises coming through the walls, unpleasant smells in the corridor) can be immediately identified and discussed.

### 5.2. Prevention of Suicide, Substance Use Disorders, and Other Risk Factors in Women with Delusional Disorder

Prevention of suicide and suicidal behavior can readily be addressed in PHPs. Prevention of social isolation is a first step and PHPs offer 7 h a day of social contact. A recent narrative review in the field provides evidence for the link between social isolation and suicidal thoughts and behaviors [68]. Both the objective condition (living alone) and the subjective feeling of being alone (loneliness) have been strongly associated with suicidal outcomes, particularly with suicidal ideation and attempts. Learning to live amicably with others is particularly important for patients with DD who generally mistrust others and are susceptible to delusional elaboration of others' motives. The setting of a PHP is in many ways ideal for bringing such suspicions to the foreground and addressing them openly. Table 4 shows potential treatment strategies to reduce both substance consumption and suicide risk.

**Table 4.** Management of substance use disorders and risk of suicide in PHPs for DD.

| Moderator | Potential Intervention | Mediator | Potential Intervention | Behaviors |
|---|---|---|---|---|
| Substance use disorders Social isolation | Psychotherapy Pharmacological treatment Psychosocial intervention | Hostility | Antipsychotics Antidepressants (when needed) Psychotherapy | Aggressive behaviors |
| | | Aggressivity | | |
| | | Impulsivity | | |
| | | Depressive symptoms | | Suicide attempts |
| | | Paranoid symptoms | | |

Sensory impairment is a risk factor for DD and also contributes to increasing rates of suicidal ideation and behavior because of its effect on the ability to socialize [68,69]. Blindness and deafness are increasing problems as one grows older, and patients with DD tend to be older individuals. PHPs can, perhaps more easily than other settings, offer alternate forms of communication [70]. The environment can be made easy to navigate for the blind, and one or two staff members can be sent for training in the use of sign language. Special equipment can facilitate participation in individual and group therapies for individuals with sensory impairments [71,72].

Persons with comorbid depressive symptoms show high rates of suicidal ideation and suicidal behavior [1]. Interventions at the medical and psychiatric nursing level to prevent suicide are focused on the identification and effective treatment of depressive symptoms. Women with DD attending PHP can be readily monitored during individual and group therapies with the aim of identifying psychiatric comorbidity. CBT for depression has proven helpful for patients with DD, mostly women [72].

Hayashi and collaborators reported the case of a 42-year-old woman with DD somatic type (delusions of infestation) [73]. The patient presented severe comorbid depression that responded to pharmacotherapy with the antidepressant, paroxetine. The monitoring of symptoms and response to pharmaceuticals is easier in PHPs than in hospitals because PHP staff is in almost constant contact with patients.

As discussed earlier, comorbid substance use disorders have been found to increase suicide risk in people suffering from mental illnesses [8,74]. PHPs have proven to be good settings in which to address substance abuse, as shown by a study conducted with veterans [75].

As mentioned, social isolation and problems in sensory processing may be risk factors for DD and for psychotic exacerbation [76]. In the case of induced DD (folie à deux), Mouchet-Mages et al. described two isolated women with a shared delusion who needed to be separated from each other in order for treatment to be effective [77]. PHPs provide partial separation for enmeshed family members as well as exposure to fresh social perspectives.

### 5.3. Coordination and Cooperation

PHPs fulfill the WHO mandate for coordination and cooperation by bridging the gap between outpatient and hospital services and by employing multidisciplinary staff.

For many decades, day treatment centers have been considered an alternative to rehospitalization for chronic psychiatric patients, such as those suffering from psychotic illnesses [78]. Day treatment centers avoided the disruption induced by a hospital stay and also assisted post-discharge patients returning to community life [78]. Since early times, different models of partial care have gradually evolved [79].

Thornicroft and Tansella [80] conducted a systematic review of modern mental health services and concluded that areas with low levels of resources need to focus on improving primary care; areas with medium resources need to provide outpatient clinics, community mental health teams, acute care and sheltered employment, whereas high-resource areas need to also offer early intervention programs and alternatives to inpatient care, which include occupational and rehabilitative services. Child and adolescent mental health care services in the community have been proposed, but high-quality evidence for their effectiveness is lacking [81], whereas, for adult community care, the evidence of effectiveness is robust [82]. Community settings and PHPs have been touted as readily accepting new perspectives such as early intervention, collaboration with families, recovery-oriented care, shared decision making, and recruitment of peer counselors—ideas that have not fully permeated in-hospital care [83–85].

With regard to patients suffering from psychosis, the model lends itself to whole person care, with emphasis not only on physical and mental health but also on occupational, recreational, and psychosocial determinants of health and quality of life. In the context of schizophrenia, there has been an emphasis on integrating supported education and supported employment into health services [86]. Recently, Humensky and collaborators investigated personnel service time requirements for supported employment and educational services through chart records from 42 individuals with a first episode of psychosis [87]. They concluded that existing financial structures could support the full range of such services. Patients with DD often have a good record of past employment, which should make this task easier than for schizophrenia.

A cross-sectional study of patients with DD investigated prevalence in two socially differentiated neighborhoods of Barcelona, Spain [88]. The study consisted of all cases recorded between 1983 and 2000 in the Case Registry of a Mental Health Unit serving two neighborhoods, La Verneda and La Mina, with 103,615 inhabitants and a high level of accessibility. Community service provision in diverse neighborhoods could prove to be excellent settings for research into the environmental determinants of delusion formation. Table 5 summarizes the main characteristics of the Barcelona study.

**Table 5.** Prevalence of DD in two socially different neighborhoods and relevance of community-based mental health models.

| Aims | | |
|---|---|---|
| To determine and compare the prevalence of DD in two neighborhoods of Barcelona, Spain (La Mina, Verneda) and describe psychosocial risk factors | | |
| **Methods** | | |
| Cross-sectional study of cases of DD included in the electronic Case Registry of the La Verneda–La Mina Community Mental Health Unit. | | |
| **Results** | | |
| | Total Sample | La Verneda | La Mina |
| Cases of DD | N = 209 | N = 145 | N = 64 |
| Prevalence of DD | 20.17/10,000 inhabitants | 18.13/10,000 inhabitants | 27/10,000 inhabitants |
| Employment Situation | Inactive: 115 (62.2%) Active: 70 (37.8%) | Inactive: 75 (60.0%) Active: 51 (40%) | Inactive: 40 (78.8%) Active: 19 (32.2%) |
| **Conclusions** | | |
| Prevalence of DD in this community-based sample is higher than the prevalence reported in hospital-based studies | | |
| Prevalence of DD is higher in neighborhoods with high frequency of psychosocial and socioeconomic risk factors | | |

Coordination and cooperation among health professionals is recommended in community based mental health teams, including collaboration with other medical services. Guàrdia and collaborators [89] reported the case of a 46-year-old woman with DD who developed a poor clinical response and unexpected adverse effects when taking a low dose of an antipsychotic drug, risperidone. Dose changes and a switch to a new antipsychotic drug did not improve the adverse events or the poor response. Monitoring the blood level of the antipsychotic drugs given to the patient (easily done in a PHP) solved the problem because it showed the patient to be a poor metabolizer of certain drugs [90]. Once she was prescribed drugs that she could metabolize, her condition improved dramatically.

*5.4. Training*

WHO recommends increased professional training for mental health personnel. Training professionals to develop specialized skills for addressing the needs of women with DD can be done on the job. Some of the required skills can be taught in school but others will require exposure to patients who come for treatment. PHPs can be used as training sites for trainees in nursing, psychology, medicine, occupational therapy, and social work. Training in the principle of shared decision making [91] is much easier in PHPs than in traditional hospital settings [92].

*5.5. Research*

Mental health research is one of the objectives of WHO's Mental Health Action Plan [12]. It is one of the critical ingredients for mental health planning and evaluation. PHPs are potentially useful settings for the clinical assessment of symptoms and of the efficacy of treatments. Psychometric instruments can be used to routinely collect data and evaluate response to pharmacological or non-pharmacological treatments. As outlined earlier, antecedents to delusion formation can also be investigated more easily than in hospital settings. One example is the investigation into the role of traumatic experience in the development of delusions [93].

Research in PHPs is easily integrated into ongoing programming. For instance, a focus on women with DD can rigorously investigate the many areas that differentiate

men and women—response to pharmacological agents or CBT, hormonal effects, domestic relationships, parental relationships, therapeutic relationships, physical and psychiatric comorbidities, substance abuse, employment opportunities, delusional content, aggressive and suicidal tendencies [94].

## 6. Discussion and Conclusions: Planning the Future

The World Health Organization (WHO) elaborated a 7-year Mental Health Action Plan in 2013 and recommended the integration of health and social care services in community-based settings, as well as the implementation of strategies for health promotion and prevention of illness. It also vigorously advocated for increased research and training for health personnel [12].

Patients with DD have been widely perceived by health professionals as suffering from a difficult-to-treat and treatment-resistant disorder [1]. Part of the difficulty may result from failing to appreciate gender-specific needs within this disorder. For instance, recent research points to a significant association between affective symptoms and delusions in patients with DD [11], more frequent in women than in men, suggesting the need to develop gender-specific interventions [3,11]. Physical health comorbidities, such as autoimmune diseases, cancers of the reproductive tracts, sleep disturbances, cardiovascular disease and dementia, also differ markedly in women and men.

In this review, we highlight partial hospitalization programs for DD, with a special focus on the needs of women because it is often taken for granted that women can be effectively treated in exactly the same way as men. We suggest that partial hospitalization programs (PHP) are in many ways ideal settings for carrying out the WHO recommendations. We have pointed out that the core strategies advocated by the WHO Mental Health Action Plan (2013–2020) can be applied in PHP settings for patients with DD [12]. The relevant themes are health promotion and prevention of mental health disorders, improvement of quality, equity and continuity of care, integration of health and social models of care, and appropriate training for staff [12].

PHPs are served by mental health staff with different training backgrounds, which provides flexibility in offering a variety of programs to the difficult-to-serve DD population. Diverse preventive and treatment services can be made available, many of which have shown proven efficacy. WHO supports early prevention that includes improvement in the social and economic circumstances of the population living in the geographic district of the PHP, and the monitoring of young people's psychological, educational, and employment needs while sheltering them, wherever possible, from adverse exposures and events [12]. Workplace interventions and post-episode reintegration into employment are vital [42,43]. While most patients with DD are able to work, they frequently, as a result of their delusional symptoms, experience interpersonal difficulties which require psychological intervention [43]. This, in our view, can be accomplished more readily in the context of partial hospitalization than in either hospital or outpatient settings. Substance abuse and depressive symptoms seen in DD are known potential risk factors for suicide. Prevention of suicide is not only underscored in the WHO 2013 document [12]; it is also eminently feasible in personalized and flexible programs such as those available in PHPs [46,47].

PHPs are a community bridge between hospital and outpatient services and can quickly adapt to the specific needs of the patients they serve. Outcomes of day treatment centers have shown evidence of effectiveness as an alternative to hospital admission and rehospitalization in psychiatric patients [78–80]. What seems to be crucial is ensuring the provision of best practice within PHPs [82]. This includes tailoring the program to address specific needs, such as the needs of women, [15] by forging collaborations with other medical specialties and with social agencies. An extra benefit is that PHPs can also serve as settings for professional training and research. They are not a panacea. They cannot serve acutely ill, suicidal or violent patients [95]. Nevertheless, they form a vital part of the safety net for psychiatric patients.

**Author Contributions:** A.G.-R., A.A., and A.G. were involved in the electronic search and selection of papers and wrote the first draft of the manuscript. R.P., J.A.M., and J.L. supported data and collaborated in the first draft of the manuscript. J.A.M., D.J.P., and J.L. critically edited the manuscript. M.V.S. edited the paper and supervised the review. All authors have read and agreed to the published version of the manuscript.

**Funding:** This research did not receive any specific grant from funding agencies in the public, commercial, or not-for-profit sectors.

**Institutional Review Board Statement:** Not applicable.

**Informed Consent Statement:** Not applicable.

**Data Availability Statement:** The data presented in this review are available on request from the corresponding author.

**Acknowledgments:** J.L. received an Intensification of the Research Activity Grant (SLT006/17/00012) from the Health Department of the Generalitat de Catalunya.

**Conflicts of Interest:** J.L. received honoraria for lectures or advisory board membership from Janssen, Otsuka, Lundbeck, and Angelini. A.G.R., A.A., and A.G.D. received honoraria or were paid for travels from Janssen, Otsuka, Lundbeck, and Angellini.

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
