# Peer review of "Psychiatric Partial Hospitalization Programs: Following World Health Organization Guidelines with a Special Focus on Women with Delusional Disorder"

_women, doi:10.3390/women1020008_

Round 1
Reviewer 1 Report
Dear Editor,
The article presents a narrative-based review of the progress in out-of-hospital mental health treatment models and partial hospitalization programs (PHP) for patients with delusional disorders (DD). The review focuses on the relationship between WHO’s Mental Health Action Plan initiated in 2013 and its success in treating women with DD through PHP. The article concludes with the authors’ suggestions on how current practices should be improved upon.
The manuscript is extremely well-written and the key messages clearly delivered. I recommend the article for immediate publication but with a couple of humble suggestions:
- Given the depth and breadth of the information presented, a general reader would greatly appreciate a flow chart or schematics summarizing (i) the authors' narrative flow of the review, (ii) the current state of the integration between various community-based treatment models and PHP, and/or (iii) the authors' suggested next steps and their interplay with regards to the different (WHO) objectives.
- There are many important statistical facts mentioned in the article which would dramatically benefit from a reproduction of the original data or charts. These include (but not limited to) facts around (1) correlation between substance abuse and suicide rates, (2) DD prevalence in various neighborhoods, (3) current treatments of DD and their associated efficacies, etc.
Author Response
REVIEWER 1
All reviewer responses in the text are in yellow.
The manuscript is extremely well-written and the key messages clearly delivered. I recommend the article for immediate publication but with a couple of humble suggestions:
1.Given the depth and breadth of the information presented, a general reader would greatly appreciate a flow chart or schematics summarizing (i) the authors' narrative flow of the review, (ii) the current state of the integration between various community-based treatment models and PHP, and/or (iii) the authors' suggested next steps and their interplay with regards to the different (WHO) objectives.
We entirely agree with Reviewer 1 that a flow chart would be helpful to understand the screening and selection processes. Figure 1 presents the methodological procedures.
At the end of the section 3 (Community based mental health models), we have added a paragraph of information about the integration between community-based treatment models and partial hospitalization programs.
Furthermore, at the end of the manuscript, we have added a discussion about next steps to be done in the interplay between community-based mental health models and partial hospitalization programs with special emphasis on family-based integrated care for complex illness (integrating family members into treatment plans and procedures) and targeting quality of life.
2.There are many important statistical facts mentioned in the article which would dramatically benefit from a reproduction of the original data or charts. These include (but not limited to) facts around (1) correlation between substance abuse and suicide rates, (2) DD prevalence in various neighborhoods, (3) current treatments of DD and their associated efficacies, etc.
We agree with Reviewer 1 that the correlation between substance use disorders and suicide risk should be better presented. Table 4 addresses potential treatment strategies to reduce both substance consumption and suicide risk.
The paper about the prevalence of DD in various neighbourhoods has been summarized in Table 5. We agree with Reviewer 1 that this would be of interest to readers to better clarify the role of social components in the frequency and occurrence of DD.
We entirely agree that treatments should be specifically addressed. In section 5.5. about research into partial hospitalization programs, we have added a paragraph highlighting the current evidence for pharmacological and non-pharmacological treatments for DD. PHPs can serve as sites at which new trials and treatment strategies can be designed and conducted.
Reviewer 2 Report
The topic of interest PHP is well addressed and conclusive, but the title makes a statement (focus on women) that is not met, neither in the abstract nor in the body text until section 5.
Lines 1-2 and 23-35. The authors point at scenarios that are flexible and have advantages, but not at specific and thus differential aspects that can be considered gender-based programs. That is, the same abstract would apply if one would refer to mental health and PHPs focused on race X or nationality Y or sex/gender Z.
Maybe the goal of the present work, in its present form, should be to summarize / review 'How 2013-2020 WHO guidelines have changed PHP for DD, with special focus on Women' . That would raise expectations that will be satisfied in 4 of the 5 sections, and the last one (the 5th) will focus on women without the feeling of being out of focus.
Lines 23-26. Seem to address our attention to the MH action plan more than to the goals to focus on women's health. The authors should short this part, focus the abstract content on their target (women) and gain some lines to propose their model of PH that responds to the health and social needs of women.
Lines 36. Consider adding the keyword 'gender-medicine'
Lines 38- The introduction starts with statements that define patients in a sex/gender dichotomic, binary order, which is not accurate nor should be accepted according to WHO, and UN. Since a review is an integrative work of literature, there's room to accept that the analysis uses the classical binary code for sociodemographic data, but the authors do not have to carry on the old structures in this respect. Check the document, while in those referring to previous work, the men/women references are correct as it refers to what was measured at that time (i.e. 66-67), take care that in the general comments (new thinking) the dichotomy is not imposed: just omit men and women to avoid binary when the sentence does not require it (i.e. line 41), just mention one of the sexes/gender (i.e.49) or refer to 'genders', are some of the easy ways to solve this problem.
Line 53. The authors refer to 'sample' as a factor that influences prevalence and incidence. However, the example provided (prison), as many others, also involve specific 'environmental factors' know to exert or to be triggers for organic and mental health problems.
Lines 89-95. Since, as referred by the authors in several paragraphs of the introduction, the guidelines of WHO are driving the goals of the review, but the present work is presented as focus on women, the authors should also include the last WHO recommendations towards women or provide a more specific description in line 108.
Also, please, note that in the current field of research, mental health and gender are two goals that make the scenario be under-considered in a double manner. This is important to be highlighted since, unfortunately, the impact is synergistic.
Line 104-110. If the aim is to focus in women, how is it that women are not included per se as a keyword in the search?
Line 112- 323 Sections 3. Table 1. Section 4. Table 2. etc are five pages not focused on women, but the focus is MH. The reader needs to wait until line 258, line 261. and more specifically until section 5.1. to hear about Health promotion in women with DD.
Minor
Line 77. …evidence base [2]. The (punctuation is missing)
Author Response
All reviewer responses in the text are in yellow.
1.The topic of interest PHP is well addressed and conclusive, but the title makes a statement (focus on women) that is not met, neither in the abstract nor in the body text until section 5.
We have corrected these omissions.
2.Lines 1-2 and 23-35. The authors point at scenarios that are flexible and have advantages, but not at specific and thus differential aspects that can be considered gender-based programs. That is, the same abstract would apply if one would refer to mental health and PHPs focused on race X or nationality Y or sex/gender Z.
We entirely agree with Reviewer 2 that the issue of gender was not sufficiently emphasized.
We have now clarified the relevance of this paper to the study of gender differences.
3.Maybe the goal of the present work, in its present form, should be to summarize / review 'How 2013-2020 WHO guidelines have changed PHP for DD, with special focus on Women' .That would raise expectations that will be satisfied in 4 of the 5 sections, and the last one (the 5th) will focus on women without the feeling of being out of focus.
We agree. We have rephrased the goals/aims of the paper; we have included community treatment approaches to patients with DD and have proposed a model of partial hospitalization compatible with WHO guidelines, with a focus on women. This is now included at the end of the introduction section.
4.Lines 23-26. Seem to address our attention to the MH action plan more than to the goals to focus on women's health. The authors should short this part, focus the abstract content on their target (women) and gain some lines to propose their model of PH that responds to the health and social needs of women.
We have listened to these recommendations and have focused the introduction on the health and social needs of women.
5.Lines 36. Consider adding the keyword 'gender-medicine'
We have included gender-specific treatment.
6.Lines 38- The introduction starts with statements that define patients in a sex/gender dichotomic, binary order, which is not accurate nor should be accepted according to WHO, and UN. Since a review is an integrative work of literature, there's room to accept that the analysis uses the classical binary code for sociodemographic data, but the authors do not have to carry on the old structures in this respect. Check the document, while in those referring to previous work, the men/women references are correct as it refers to what was measured at that time (i.e. 66-67), take care that in the general comments (new thinking) the dichotomy is not imposed: just omit men and women to avoid binary when the sentence does not require it (i.e. line 41), just mention one of the sexes/gender (i.e.49) or refer to 'genders', are some of the easy ways to solve this problem.
We appreciate this recommendation. We have tried to revise the manuscript accordingly but have found the use of non-binary language to be difficult to insert into the text.
7.Line 53. The authors refer to 'sample' as a factor that influences prevalence and incidence. However, the example provided (prison), as many others, also involve specific 'environmental factors' know to exert or to be triggers for organic and mental health problems.
We have changed the wording in order to be more precise.
8.Lines 89-95. Since, as referred by the authors in several paragraphs of the introduction, the guidelines of WHO are driving the goals of the review, but the present work is presented as focus on women, the authors should also include the last WHO recommendations towards women or provide a more specific description in line 108.
We have added some more specific details about WHO recommendations re women. All changes and implementations can be found in section 4.
9.Also, please, note that in the current field of research, mental health and gender are two goals that make the scenario be under-considered in a double manner. This is important to be highlighted since, unfortunately, the impact is synergistic.
Thank you. We have now highlighted this double jeopardy.
10.Line 104-110. If the aim is to focus in women, how is it that women are not included per se as a keyword in the search?
We chose, perhaps mistakenly, to undertake a general search strategy with a subsequent identification of studies related to gender.
11.Line 112- 323 Sections 3. Table 1. Section 4. Table 2. etc are five pages not focused on women, but the focus is MH. The reader needs to wait until line 258, line 261. and more specifically until section 5.1. to hear about Health promotion in women with DD.
We have rearranged paragraphs so that the focus on women is clear earlier in the paper.
12.Line 77. …evidence base [2]. The (punctuation is missing).
We have corrected this.
Reviewer 3 Report
The article is interesting, well written and structured. The topic of health care for people with delusional disorder is important during recent times, especially during the covid pandemic.
References regarding the Community Partial Hospitalization Therapy and Rehabilitation, which is the main topic of this article, are insufficient, as compared to the total number (78 references). Consider to retrieve and analyze more articles and books available from international literature, Focused on Community Mental Health Therapy CMHT and Partial Hospitalization Programs PHP.
Avoid references in different languages, many international readers can not use them.
In this review article the authors must add some specific results from real study cases, with clear obtained results, (can be of the authors, or reported in literature). The case reports must be of women (because the Women journal aims are “women’s health, social determinants of health, and healthcare research”).
Based on reported cases, give a frank estimation of the strengths and weaknesses of CMHTs and PHPs.
Author Response
All reviewer responses in the text are in yellow.
1.References regarding the Community Partial Hospitalization Therapy and Rehabilitation, which is the main topic of this article, are insufficient, as compared to the total number (78 references). Consider to retrieve and analyze more articles and books available from international literature, Focused on Community Mental Health Therapy CMHT and Partial Hospitalization Programs PHP.
We retrieved and added references to Community Mental Health treatments and partial hospitalization programs according to the suggestions of Reviewer 3. Most addressed psychosis in general; we have tried to translate the evidence that applies to women with DD.
2.Avoid references in different languages, many international readers can not use them.
This used to be true but automatic translation is now readily available and many international readers prefer to read the occasional article in a language more familiar to them than English. We have, nevertheless, limited references to articles written in languages other than English.
3.In this review article the authors must add some specific results from real study cases, with clear obtained results, (can be of the authors, or reported in literature). The case reports must be of women (because the Women journal aims are “women’s health, social determinants of health, and healthcare research”).
We have now added some brief examples of published cases of women with DD.
4.Based on reported cases, give a frank estimation of the strengths and weaknesses of CMHTs and PHPs.
In the discussion section, we have addressed the strengths and weaknesses of partial hospitalization programs and CMHTs in the context of women with DD.
Round 2
Reviewer 2 Report
The authors have taken in consideration all questions and implemented those feasible and highlighted some potential aspects that were hidden but were part of its potential.
In my opinion, the Ms is now more close to the scope of this journal, provides valuable information for clinical approach on desilusional disorder focusing on woman and WHO recommendations,
From my part it's suitable for publication in its current form.
Reviewer 3 Report
The authors fulfilled the most of this reviewer's comments.
The article is very much improved and can be accepted as is.